# Preliminary Effectiveness of a Strategy to Promote Healthy Lifestyle Habits in Schoolchildren

**DOI:** 10.3390/children9091402

**Published:** 2022-09-16

**Authors:** Gloria Carvajal-Carrascal, Tania Catalina Chinchilla-Salcedo, César Augusto Niño-Hernández, Paola Sarmiento-González, Angélica María Ospina-Romero, Beatriz Sánchez-Herrera

**Affiliations:** School of Nursing and Rehabilitation, Universidad de La Sabana, Chía 250001, Colombia

**Keywords:** nursing methodological research, school health, health policy, health promotion, community-institution relations

## Abstract

Purpose: To measure the preliminary effectiveness of a strategy to promote healthy lifestyle habits in schoolchildren, aged 6 to 12 years, living in the Andean region of Colombia, 2018–2021. Design and Methods: This is a Nursing Methodological Research, developed in phases: (1) Context and schoolchildren characterization; (2) Strategy design guided by the Whittemore and Grey criteria and the Bronfenbrenner ecological conceptual model; (3) Strategy validation with 11 experts; (4) Trial to evaluate preliminary effectiveness. We applied the strategy in seven different schools with the educational community including 955 schoolchildren between 6 and 12 years of age, 551 parents, 130 teachers and 7 members of the food staff. Results: Our health promotion strategy “Prosalud” has five components: citizenship feeding and nutrition, physical activity, and rest; environment protection, and directing one’s own life. It includes all the participants of the educational community. Conclusions: According to experts, the health promotion strategy “Prosalud” is valid to help promoting healthy lifestyle habits among schoolchildren. This strategy demonstrates its preliminary effectiveness in a group of school children from 6 to 12 years of age, their parents, teachers, and school food staff, in the Andean region of Colombia.

## 1. Introduction

The increase in chronic disease worldwide has generated new priorities for health systems. If they do not address this situation, the consequences could be catastrophic for the quality of life of people and for sustainable development, especially in middle- and low-income countries [1].

The figures for chronic health problems are not encouraging in Latin America and Colombia, despite efforts to promote health [2,3,4]. Therefore, it is necessary to develop and validate new strategies that have a positive impact on the social determinants of health and help guarantee the right to life and well-being of people, especially children and adolescents [5].

The guidelines for working on health promotion in children and adolescents state that it is important for there to be collaboration between health and education professionals, and institutions, in the school context [6]. These efforts to promote the knowledge, attitude and behavior necessary to ensure a healthy lifestyle are complex but can undoubtedly be more effective, especially during childhood and adolescence [7].

Recent advances in the field of school health promotion show the relevance of ecological models to design programs [8]. Different authors suggest linking all members of the educational community in these health promotion strategies. This includes schoolchildren [9,10], their parents [11], teachers [12] and other members of their environment [13].

School health promotion developments have focused on five main areas with evidence of significant progress. These are the strengthening of coexistence and citizenship skills [10,12,14]; adequate nutrition, hydration, and food intake [13,15]; physical activity and rest [11,16,17]; relationship of schoolchildren with the environment and the promotion of pro-environmental behaviors [18]; and the strengthening of mental health through the identification of a life project and good company, which is associated with better academic performance of schoolchildren [19].

The situation of the school population in the Andean region of Colombia shows negative indicators in relation to their nutritional status, school desertion, consumption of psycho-active substances, increased violence and abuse, early pregnancies, and contexts with high pollution [20]. This complex situation must be addressed to improve the quality of life of children and adolescents and reduce the chances of an increase in chronic disease and its consequences. The objective of this study is therefore to develop a strategy to promote healthy life habits in school children aged 6 to 12 years, living in the Andean region of Colombia, and validate its preliminary effectiveness. This strategy is based on the best available evidence and maintains an ecological approach to recognize and generate synergies between the members of the educational community and the resources available to promote their health [21].

## 2. Materials and Methods

This is a Methodological Nursing Research aimed to design and validate a strategy to promote healthy lifestyle habits in schoolchildren from 6 to 12 years of age living in the Andean region of Colombia. This research is part of the school’s health promotion project called “Prosalud—US” 2018–2021. Its development considered ethical and environmental aspects and was endorsed by the Research Ethics Committee of the Institution (Approval Act No. 0030518 of 5 March 2018).

We developed this study in four sequential phases. The first was context and schoolchildren characterization. We included a description of the health and care conditions of schoolchildren in the Andean region of Colombia by an exhaustive review of their health context from secondary sources. Then we conducted the identification of these school children care conditions guided with the GCPC-UN-P characterization survey for care. This tool was validated in Spanish [22] and adapted for the school population. This tool includes a sociodemographic profile, the perception of caring practices and the available support for them, and the use of information and communication technologies (ICT) as an element to support care practices. Before the interview, we obtained an informed consent from the parents and an informed assent from the minors. For this diagnosis, we intentionally selected a sample of 95 schoolchildren aged 6 to 12 years. Our sample included a representation from different municipalities of the Andean region and from rural, urban, public and private institutions. We processed the data in Excel and used descriptive statistical tools for analysis.

### 2.1. Strategy Design

After seeking for the best available evidence, we defined a new strategy to promote healthy lifestyles in schoolchildren under the Whittemore and Grey criteria for systematic health interventions [22]. We received three face-to-face consultations with the author, in which we verified the standards for the development of the intervention strategy and their strict compliance. We also guided our strategy by the Bronfenbrenner Ecological Model [23]. This conceptual model indicates that the promotion of healthy life habits in school children should consider all the people within the systems in which these children interact in their daily lives. For our study these included their parents, teachers and the school food staff.

### 2.2. Strategy Expert Validation

The strategy was validated by an expert focus group that included 11 school principals with more than 5 years of experience in the management of school populations in the Andean region of Colombia. For this purpose, we made a presentation of the strategy that included the problems detected in the school population, its characteristics and context; the objective; the participants; the contents and domains; the dosage and duration; the resources required and the form and place of delivery; and the expected proximal, primary, secondary and distal results. We asked the experts for their evaluation in terms of clarity, relevance, pertinence and sufficiency of each component of the strategy and of the complete strategy, as well as their suggestions to correct or complement our proposal.

### 2.3. Trial to Evaluate Preliminary Effectiveness

After the strategy was validated with experts, we applied it with the educational community of seven different schools of the Andean region. This application included 955 schoolchildren between 6 and 12 years of age, 551 parents, 130 teachers and 7 members of the school food staff. During this phase all the institutions boards, parents, teachers and food staff signed an informed consent; all the schoolchildren signed an informed assent. To measure the strategy preliminary effectiveness, we evaluated each participant before and after the intervention under the Prosalud guidelines Carvajal et al. [24]. Based on this Prosalud guide, we elaborated the HM-HVS- ESC (measurement tool for healthy lifestyle habits in schoolchildren), in which, through a methodological study, we assessed its comprehensibility with 21 children aged 6 to 12 years. We defined its face and content validity with 8 experts by calculating the content validity ratio (CVR) of each item and the global content validity index (CVI) under Lawshe’s parameters modified by Tristan [25]. This allowed us to guarantee the clarity, sufficiency, pertinence and relevance of the tool. Based on these parameters, the agreement index among the experts who validated the tool was 78.3 and the CVI was 0.82. However, there is no factor analysis of the tool. These guidelines include the knowledge, attitude and practice towards healthy lifestyle of children, parents, teachers and food staff. The period between measurements was 6 months. For the analysis, we standardized all the responses to 100, to be able to make interpretations and comparisons. We classified values of 80 points or more as high; values from 60 to 79 as medium; and values of 59 or less as insufficient. For the analysis we compared the results, using SPSS version 22, by establishing frequency tables and response percentages before and after the intervention to determine whether the strategy had a positive, negative or neutral effect on keeping healthy habits. We made intragroup and intergroup comparisons to establish the preliminary effect of the strategy, selecting two dimensions of the strategy: feeding and nutrition, and physical activity and rest. We used the signed rank test for the intragroup analyses (pretest–post-test) and the Mann–Whitney U test for the intergroup analyses. The statistical significance value was *p* < 0.05.

## 3. Results

In the review of the context and the caring practices of schoolchildren, we found that, in these 11 municipalities of the Andean region, there are 115,128 schoolchildren, 54.9% of whom are enrolled in public institutions and 41.5% in private ones. The regional quality-of-life observatory Sabana Centro Cómo Vamos [20] reported a high incidence of pregnancies, consumption of harmful substances, malnutrition due to deficit or excess of foods, school dropout, violence and self-harm, as well as lack of knowledge of health care practices for this schoolchildren population. However, this region has resources, information and support to work on behalf of children and adolescents as it is demonstrated by s successful experiences supported by the State, private organizations and academia [26]. We found a regional strength with the private enterprise implementation of the strategy United for Healthy Children. This strategy has national and international validation, implementation and socialization [27]. Based on its positive evaluation and prior intersectoral endorsement, we selected and integrated two of its components: nutrition and physical activity.

These group of schoolchildren consisted of 54.5% boys and 45.3% girls, 99% with high levels of physical functionality and 97.8% with an adequate cognitive performance. Of these children, 11.6% reported diseases including allergies, respiratory, neurological or ocular alterations. A total of 29.5% come from a municipality in the Andean region other than the municipality in which they study and 17.8% come from other places outside the region. Among the children, 69.5% live in rural areas, with 82.8% living in the lower-middle and lower strata, 2 to 3 out of 6 possible. Although the majority only study, 25.3% reported they conduct other activities such as helping around the house or playing sports. Regarding religion, 96.8% say they belong to a religion, and 84.2% report being Catholics; 40% report that their level of religious commitment is high.

These schoolchildren report physical and psychological well-being in 81.1% of cases, social well-being in 83.2% of cases, and spiritual well-being in 61.1%. A total of 96.8% of the schoolchildren are cared for by their families: 65.3% by parents, 21% by grandparents, 8.4% by siblings and 2.1% by aunts and uncles. Moreover, 69.5% perceive that they are cared for more than 12 h daily.

Regarding their perception of support, 67.4% perceive having psychological support and 52.0% rate it as adequate; 64.6% perceive having family support and 86.3% consider it adequate; 94.8% perceive having religious support and 41.1% consider it adequate; 97.9% perceive having economic support and 79% consider it adequate; 96.9% perceive having social support and 69.4% rate it as adequate. Of the children, 87.4% do not perceive themselves to be a burden for the family; however, 3.1% perceive themselves to be a high burden and 2.1% an extremely high burden for the family.

Pertaining to their appropriation of ICTs for their care practices, we found a medium level of adoption for television and internet, and a low level for radio, computer and telephone.

The proposed health promotion for schoolchildren strategy is call “Prosalud” as an acronym in Spanish: PRO, which stands for “promoción” (promotion in English) and “SALUD” (health in English) and refers to knowing and applying citizenship competencies; eating well; achieving adequate physical activity and rest; protecting the environment and directing one’s own life. This strategy includes the definition of the problem that requires intervention, the persons to whom the intervention is addressed, the intervention pathway, how the intervention is developed and its expected outcomes. It also includes the guidelines to work with schoolchildren, parents, teachers and school food service personnel, as well as the guidelines for working with policy makers and with the media (Table 1).

### 3.1. Strategy Expert Validation

During the validation, the participating experts found the strategy to be complete, clear and pertinent. Their opinions unanimously agree about the importance of this approach within the school community in response to the major school children health problems. The experts suggested that the strategy should also be shared with schoolchildren, parents, teachers and service personnel, which we did prior to its implementation.

### 3.2. Trial to Evaluate Preliminary Effectiveness

The field implementation of the strategy with the educational community reflected that the strategy was easily understood, well accepted and preliminarily effective in improving the knowledge, attitude and behavior of the educational community to promote the health of schoolchildren. This first measure showed a preliminary positive effect with changes in the knowledge, attitude and behavior of children, namely, 64.8–65.9%, maintaining them in a medium level; their parents, 68.9–69.7%, also maintaining them in a medium level; and the school food service personnel, 53.4%–64.7%, changing from a low to a medium level. However, the teachers did not show improvement in this period with the strategy implementation, although they maintain their average in the higher level, 80.6–79.9% (see Table 2).

The pretest—post-test intragroup analysis showed statistically significant differences for the Feeding and nutrition dimension in the following components: knowledge about the benefits of water consumption (*p* = 0.045), adequate food portion size (*p* = 0.027), benefit of food groups (*p* = 0.002), and benefits of eating and cooking as a family (*p* = 0.033). Likewise, statistically significant differences were observed in the component benefits of physical activity (*p* = 0.029) which corresponds to the dimension of physical activity and rest.

There were significant differences in the nutrition dimension within the components when comparing the two selected dimensions between groups of schoolchildren. They were in the benefits of water consumption (*p* = 0.014), benefits of food groups (*p* = 0.007) and benefits of eating and cooking as a family (*p* = 0.000). In the physical activity dimension, they were in the components of frequent practice of physical activity and sport (*p* = 0.004), benefits of physical activity (*p* = 0.004) and time spent doing sport (*p* = 0.003).

## 4. Discussion

This health promotion strategy is a response to the universal call to promote the health of the school-age children and the general population, especially in low- and middle-income countries. The figures of malnutrition and sedentary lifestyles in Latin America are increasingly alarming, as reported by Campos et al. [27].

This “Prosalud” strategy considers the evidence retrieved by Nash et al. [29]. On the one hand, it employs an ecological model that integrates contexts and knowledge while maintaining its central focus on children. On the other, it has a multicomponent design that simultaneously addresses various knowledge, attitudes and behaviors that affect health. The strategy is directed towards the schoolchildren, their teachers, parents and food staff, so that all participants are supported and support schoolchildren’s daily life. Finally, this strategy also responds to the call to work with schoolchildren before adolescence [29]. It is important to highlight that although knowledge, attitude and behavior improved with the strategy, the results of the categories show non satisfactory levels, except for citizenship competencies, especially in the case of nutrition and physical activity of schoolchildren. This indicates the necessity to reinforce this strategy over time to help maintain positive habits and redirect those that are not.

Like Abrahamse (2018) [30], this strategy promoted the comprehensive, interdisciplinary and intersectoral health of schoolchildren. Even though most have socioeconomic limitations that can negatively affect their present and future development, we identified and incorporated other potential advantages like family and school management support.

In this study, we follow the trail set in Spain by Santos-Beneit (2019) [31] and respond with a contextualized strategy to modify unhealthy lifestyle habits in schoolchildren aged 6 to 11, 6 to 12 in our case, encouraging them towards healthy lifestyles supported by their parents and teachers, and hoping that it will be sustainable over time.

Our strategy includes five components: citizenship skills, eating and nutrition, physical activity and rest, environmental protection and direction of one´s own life.

In its first component, citizenship skills, our proposal encourages school children to actively participate in collective projects. Our results support that citizenship is a construction that affects all school community members. Teaching citizenship competencies, as established by Dunhill (2018) [32], incentivizes respect for others in children, as evidenced in the present study.

In our second strategy component, eating and nutrition, we agree with the findings of Barnes et al. (2017) [33], who point out that public policies have a positive impact on nutrition. Similarly, it is aware of the findings of Smith et al. (2020) [34], who establish that the family model strengthens the nutrition of children. However, it would be desirable to continue working in this field to contribute with the results to the evidence that, as indicated by Black et al. (2017) [35], still appears to be weak.

In our third component, physical activity and rest, our proposal coincides with that reported by Kliziene (2018) [36], in his program to improve physical activity levels in schoolchildren and with Finn et al. (2018) [37] who seek to integrate academic activities with physical activity. Unlike these authors, “Prosalud” incorporates adequate rest guidelines that were neither known nor practiced by the group.

In our environmental protection component, it is striking that, despite the existence of social movements worldwide to see and understand the school as a whole, the environment is poorly integrated into school health approaches, as pointed out by Sarmiento et al. (2019) [38]. “Prosalud”, in its approach of joining the environment, agrees with Gkotzos et al. (2017) [39], who seeks to integrate this component as an activity in the educational plan. In fact, the present proposal strengthens the application of scientific knowledge for environmental care and mobilizes young generations to apply the knowledge they acquired, as suggested by Bang et al. (2018) [40].

Finally, in the health promotion strategy component of directing one’s own life, strengthening helpful friendships implies greater self-awareness and strength in decision-making, which, as Markham (2019) points out, significantly affect health-related behaviors [41]. As proposed by Kitching et al. (2020) [42], “Prosalud” points out that the holistic wellbeing of the schoolchild includes a look at the links that the schoolchild has, in addition to the context and a clear conceptual approach. According to the authors, the friend selection of schoolchildren may predispose them to success or failure in their present and future.

As evidenced by the results of the preliminary evaluation of the “Prosalud” strategy, it has a positive effect. However, two considerations arise: First, although it is an important achievement to go from an overall average of 66.9% to 70.1% regarding health knowledge, attitudes, and behavior evaluation results, it must be recognized that these strategies need continuity over time to close the gap between the progress achieved and the desired scenarios to transform health for this population. The second, but equally relevant, is the commitment and motivation that teachers should have, since they did not show a change in this preliminary evaluation, but their knowledge, attitude and behavior can be definitive to modify the health habits of schoolchildren [43,44].

The “Prosalud” strategy, in addition to addressing the issues of the schoolchild’s own development, generated synergies with their context, seeking the development of leadership in the school, like Fergusson et al. (2021) [45].

We expect the “Prosalud” strategy to achieve better results withs its continuous application. In the same way Faught et al. (2018) [46] did by comprehensively addressing the habits of minors, better results will be achieved both in health and academic achievement with potential benefits in time and resources.

This makes it necessary to generate a prospective plan that articulates the different sectors of the school community, so that they modify the culture to achieve healthy lifestyle habits that can be more sustainable over time and achieves the changes we propose. The inclusion of public policy and the media in “Prosalud” could be a strategic contribution in the field.

## 5. Conclusions

The Prosalud strategy was designed to promote healthy lifestyle habits in school children between 6 and 12 years of age living in the Andean region of Colombia. This strategy is focused on the selected group of schoolchildren and responds to their characteristics by maintaining an ecological perspective that also included their parents, teachers and ser-vice personnel. The educational community was considered in its specific context and culture.

The design of the Prosalud strategy responds to the international standard for the development of health interventions. Its contents are grouped into five components: the promotion of citizenship skills, adequate food and nutrition, physical activity and rest, the promotion of pro-environmental behaviors, and the strengthening of the life project through the selection of good company.

The experts convened for the validation considered that the Prosalud’s strategy is clear, understandable, relevant and sufficient for this educational group and community.

The application of the strategy with the educational community in the selected schools showed its applicability and a positive preliminary effect on them. Statistical tests on the components of nutrition and adequate food, and physical activity and rest of schoolchildren showed statistically significant changes.

The authors accept as a limitation of the present study that, to date, there is no factor analysis of the HM-HVS-ESC tool.

These findings are applicable to schools in the Andean region of Colombia and could be useful in educational communities with similar conditions.

## Figures and Tables

**Table 1 children-09-01402-t001:** Design criteria for the health promotion strategy “Prosalud” with schoolchildren in the Andean region of Colombia, 2018–2021, under the application of the Whittemore and Grey parameters [28].

Dimension	Criteria	Description of the Strategy Prosalud-US
Problem requiring intervention	Problem to intervene in a population at specific risk	Need to strengthen habits for a healthy life in the population of children from six to twelve years of age in the Andean region of Colombia.
Person(s) to whom the intervention is addressed	Unit of analysis	Children from six to twelve years of age Parents Teachers and principals Shopkeepers or school food service personnel Local authorities representing the state on education and health issues Media
How the intervention will be developed	The contents and domains that are addressed with the intervention	Health Promotion Habits related to: Citizenship competencies (peaceful coexistence in living environments and interaction, social coexistence with participation and respect, cooperation and communication, commitment to one’s own and the common seeking collective well-being). Feeding and nutrition (water consumption, managing food portions, choosing variety and nutrition, eating and cooking as a family). Physical activity and rest (promoting exercise, physical activity and sports, avoiding sedentary behaviors, sleeping and resting adequately). Environment protection (environmental awareness in relation to water, energy consumption, waste, reuse and recycling, and transportation. Directing one’s own life (self-knowledge, decision making, positive attitude, elaboration of life projects, identifying good company and appropriate use of ICT).
	Dosage and duration	Five sessions are required, 1 for each dimension, directed to each of the population segments, with a duration of 1 h per session, with an intensity of 1 session per month for a total of five months.
	Delivery strategy	Group, face-to-face, with workshops and playful support for the child population.
	Environment or place of execution	The school, the rector’s network, the municipal government
	Resources	Qualified personnel, two people for each activity in classes of a maximum of thirty-five children and similar groups of parents, teachers, directors and shopkeepers. Direct work with local government and media. Recreational and pedagogical material according to the population segment, the content, and the strategy to be developed.
Outcomes	Proximal, primary, secondary and distal outcomes	Proximal: Improve knowledge of habits for healthy living associated with citizenship, healthy eating, adequate physical activity and rest, environment protection and one’s own self-direction. To strengthen the attitude in relation to habits for a healthy life associated with citizenship competencies, healthy eating, adequate physical activity and rest, the environment and the direction of one’s own life by strengthening good company. To promote healthy behavior in relation to health promotion habits associated with citizenship skills, healthy eating, adequate physical activity and rest, the environment and the direction of one’s own life by strengthening good company. Primary: Maintenance of a healthy behavior in relation to citizenship skills, healthy eating, adequate physical activity and rest, the environment and the direction of one’s own life by strengthening good company, until the habit is formed. Distal: Strengthen public policy by joining the strategies in place for the well-being of children in the Sabana Centro region.

**Table 2 children-09-01402-t002:** Preliminary effectiveness results of the health promotion strategy “Prosalud” with the school community in the Andean region of Colombia, 2018–2021.

Dimension	Component	Preliminary Effectiveness of the Strategy							
		School Children		Parents		Teachers		Food Staff	
		Pre	Post	Pre	Post	Pre	Post	Pre	Post
Citizenship	Recognizes respect as citizenship behavior	94.7	95.6	97.4	95.8	100	95.8	-	-
	Behavior reflects tolerance and respect	87.1	88.2	92.1	92.3	96.6	97.9	-	-
	Identifies interactions within the environment	88.5	86.7	95.2	94.4	100	100	-	-
	Identifies the value of living together in peace	94.5	92.6	94.7	95.8	98.3	100	-	-
	Recognizes the importance of being a good citizen	38.1	42.1	57.9	64.8	84.5	93.6	-	-
	Subtotal	80.6	81.0	87.5	88.6	95.9	97.5	-	-
Feeding and nutrition	Choice of healthy beverage	62.2	76.4	24.2	20.1	-	-	47.1	77.8
	Positive attitude about water consumption	49.4	48.7	63.3	62.6	84.2	86.8	-	-
	Knows daily water requirement	36.3	48.9	41.5	63.1	46.8	57.8	61.8	61.2
	Identifies importance of water consumption	75.1	76.5	71.5	70.1	64.6	75.6	69.1	72,3
	Recognizes the benefits of drinking water	62.9	63.3	-	-	80	88	70.6	81.1
	Eats a balanced diet on a daily basis	18.7	22.5	37.3	46.3	55.2	52.9	52.9	66.7
	Identifies benefits of food	34.5	42.9	37.1	37.6	68.7	68.5	64.7	77.8
	Selects appropriate portion sizes	42.2	46.1	24.3	26.2	51.6	55.6	41.2	55.6
	Knows number of meals required per day	22.4	34.8	49.9	46.6	90.4	71.4	58.8	78.8
	Organizes healthy plate	33.4	41.5	41.4	44.5	41.9	61.2	64.7	66.7
	Prepares healthy lunch box	-	-	30.3	34.9	-	-	-	-
	Positive attitude about eating with family	70.6	70.6	-	-	-	55.1	41.2	55.6
	Recognizes benefits of eating with family	71.1	72	-	-	-	85.5	70	85.6
	Washes hands before eating	78.4	77.7	-	-	93.5	83.7	82.4	88.9
	Subtotal	50.6	55.5	42.1	45.2	68.1	70.2	60.4	72.3
Physical activity and rest	Practice of physical activity, exercise or sport	94.5	95.4	-	-	-	-	-	-
	Number of days of physical activity	20.1	18.4	-	-	-	-	-	-
	Hours per day of physical activity	34.3	36.6	-	-	-	-	-	-
	Recognizes benefits of physical activity	74.5	76.3	73.6	71.3	81	83.0	73.5	74.4
	Knows physical activity requirement/day	24.2	24.8	20.5	25	22.6	22.4	29.4	55.6
	Attitude towards physical activity, exercise or sport	80	80.3	-	-	-	-	-	-
	Knows time allocation parameters	41.3	42.0	34.5	34.5	-	-	-	-
	Balances physical activity and daily rest	42.6	45.5	40.3	38.9	-	-	-	-
	Knows the importance of physical activity	-	-	68.5	67.7	64.6	59.2	47.1	66.7
	Knows the importance of doing sports	-	-	-	-	70.9	63.3	52.9	66.7
	Identifies reasons to avoid sedentary lifestyles	-	-	80.8	75.5	58.1	53.1	29.4	22.2
	Promotes physical activity	-	-	-	-	30.1	27.9	-	-
	Subtotal	51.4	52.4	53.0	52.2	54.6	51.5	46.5	57.1
Environment protection	Recognizes the importance of protecting the environment	82	76.9	74.8	78.2	91.4	97.9	-	-
	Knows recycling practices	56.6	61.8	81.6	83.8	89.7	66	-	-
	Properly handles recycling and disposal	95.9	94.4	94.2	93.7	96.6	100	-	-
	Subtotal	78.2	77.7	83.5	85.2	92.6	88.0	-	-
Leading one’s own life	Knows the importance of good company	35.1	40.7	66.3	70.4	86.2	87.2	-	-
	Recognizes the importance of making a life plan	38.9	45.2	55.8	50	91.4	89.4	-	-
	Attends responsibility with their care	73.3	68.1	85.3	83.1	79.3	87.2	-	-
	Recognizes conditions of good company	91.2	88.7	91.6	87.4	96.6	97.9	-	-
	Develops life plan	88.7	88.7	94.2	95.8	91.4	93.6	-	-
	Identifies implication of good company	90	83	94.2	95.8	100	91.4	-	-
	Understands the meaning of autonomy	25.6	24.5	61.6	57	98.3	100	-	-
	Subtotal	63.3	62.7	78.4	77.1	91.9	92.4	-	-
Total weighting of results of the strategy		64.8	65.9	68.9	69.7	80.6	79.9	53.4	64.7

## Data Availability

Data available on request due to restrictions, e.g., privacy or ethical. The data presented in this study are available on request from the corresponding author. The data are not publicly available due to the group is interesting in improving and making contacts with different organizations that are interesting in the same topics.

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
