# Peer review of "Preliminary Effectiveness of a Strategy to Promote Healthy Lifestyle Habits in Schoolchildren"

_children, 2022, doi:10.3390/children9091402_

Round 1
Reviewer 1 Report
The purpose of the study is to promote healthy life-12 style habits in schoolchildren, from 6 to 12 years of age. The authors present a Nursing Methodological Research, developed in 4 phases: 1) Context and schoolchildren characterization. 2) Strategy design guided by the Whittemore and 15 Grey criteria and the Bronfenbrenner ecological conceptual model. 3) Strategy validation with 11 16 experts. 4) Trial to evaluate preliminary effectiveness. The authors applied the strategy in seven different 17 schools with the educational community including 955 schoolchildren between 6 and 12 years of 18 age, 551 parents, 130 teachers and 7 members of the food staff.
The part of design and methodology are representative and excellent prepared. The all conclusions consistent with the evidence and arguments and are relevant to the main question that posed. It would be appropriate to include 10 more literary sources from the last 5 years. The tables are appropriate to the purpose and the results.
Author Response
The authors are grateful for the detailed review and suggestions that undoubtedly allowed us to improve the original version of the article.

Reviewer 2 Report
Dear authors, the topic studied in your manuscript on healthy lifestyle habits in schoolchildren is very interesting. However, I am concerned about conceptual and methodological aspects of your work.
1) The introduction (Literature review) is too brief and does not cover essential aspects related to the five dimensions studied: Citizenship, Food and nutrition, Physical activity and rest, Environmental protection and Self-management.
2) The reference Carvajal et al. appears to be a non-accessible document, on which the methodology of this study is based. It is necessary to provide some background as supplementary material.
3) The results analysis requires a more robust statistical treatment. As an example 3.1) to consider a confirmation about how the different items (components) explain the five factors (dimensions), a confirmatory factor analysis may be sufficient. 3.2) in the comparison between pre and post measurements ("the period between measurements was 6 months") it is necessary to analyze a statistically significant difference.
Author Response
We really appreciate the complete revision and suggestions we receive.

Round 2
Reviewer 2 Report
Dear Authors, if the dimensions do not support a confirmatory factor analysis, at least some measure of internal consistency should be reported for each dimension, to ensure that the instrument measures what is stated. If this cannot be assured, it is not possible to draw your conclusions.
Author Response

(The authors gave the same response as above.)
